# Not All Images are Worth 16x16 Words: Dynamic Transformers for Efficient Image Recognition

Yulin Wang[1][*]  Rui Huang[1][*]  Shiji Song[1]  Zeyi Huang[2]  Gao Huang[1,3][†]

[1]Department of Automation, BNRist, Tsinghua University, Beijing, China
[2]Huawei Technologies Ltd., China
[3]Beijing Academy of Artificial Intelligence, Beijing, China
{wang-yl19, hr20}@mails.tsinghua.edu.cn, huangzeyi2@huawei.com.
{shijis, gaohuang}@tsinghua.edu.cn

## Abstract

Vision Transformers (ViT) have achieved remarkable success in large-scale image recognition. They split every 2D image into a fixed number of patches, each of which is treated as a token. Generally, representing an image with more tokens would lead to higher prediction accuracy, while it also results in drastically increased computational cost. To achieve a decent trade-off between accuracy and speed, the number of tokens is empirically set to 16x16 or 14x14. In this paper, we argue that every image has its own characteristics, and ideally the token number should be conditioned on each individual input. In fact, we have observed that there exist a considerable number of "easy" images which can be accurately predicted with a mere number of 4x4 tokens, while only a small fraction of "hard" ones need a finer representation. Inspired by this phenomenon, we propose a Dynamic Transformer to automatically configure a proper number of tokens for each input image. This is achieved by cascading multiple Transformers with increasing numbers of tokens, which are sequentially activated in an adaptive fashion at test time, i.e., the inference is terminated once a sufficiently confident prediction is produced. We further design efficient feature reuse and relationship reuse mechanisms across different components of the Dynamic Transformer to reduce redundant computations. Extensive empirical results on ImageNet, CIFAR-10, and CIFAR-100 demonstrate that our method significantly outperforms the competitive baselines in terms of both theoretical computational efficiency and practical inference speed. Code and pre-trained models (based on PyTorch and MindSpore) are available at https://github.com/blackfeather-wang/Dynamic-Vision-Transformer and https://github.com/blackfeather-wang/Dynamic-Vision-Transformer-MindSpore.

## 1   Introduction

Transformers, the dominant self-attention-based models in natural language processing (NLP) [10, 40, 3], have been successfully adapted to image recognition problems [11, 55, 38, 17] recently. In particular, vision Transformers achieve state-of-the-art performance on the large scale ImageNet benchmark [9], while exhibit excellent scalability with the further growing dataset size (e.g., on JFT-300M [11]). These models split each image into a fixed number of patches and embed them into 1D tokens as inputs. Typically, representing the data using more tokens contributes to higher prediction accuracy, but leads to intensive computational cost, which grows quadratically with respect to the

---

[*]Equal contribution.
[†]Corresponding author.

35th Conference on Neural Information Processing Systems (NeurIPS 2021).

token number in self-attention blocks. For a proper trade-off between efficiency and effectiveness, existing works empirically adopt 14x14 or 16x16 tokens [11, 55].

In this paper, we argue that it may not be optimal to treat all samples with the same number of tokens. In fact, there exist considerable variations among different images (e.g., contents, scales of objects, backgrounds, etc.). Therefore, the number of representative tokens should ideally be configured specifically for each input. This issue is critical for the computational efficiency of the models. For example, we train a T2T-ViT-12

Table 1: Accuracy and computational cost of T2T-ViT-12 with different token numbers on ImageNet.

| # of Tokens | 14x14 | 7x7 | 4x4 |
|---|---|---|---|
| Accuracy | 76.7% | 70.3% | 60.8% |
| FLOPs | 1.78G | 0.47G | 0.21G |

[55] with varying token numbers, and report the corresponding accuracy and FLOPs in Table 1. One can observe that adopting the officially recommended 14x14 tokens only correctly recognizes ~15.9% (76.7% v.s. 60.8%) more test samples compared to that of using 4x4 tokens, while increases the computational cost by 8.5x (1.78G v.s. 0.21G). In other words, computational resources are wasted on applying the unnecessary 14x14 tokens to many "easy" images for which 4x4 tokens are sufficient.

Motivated by this observation, we propose a novel *Dynamic Vision Transformer* (DVT) framework, aiming to automatically configure a decent token number conditioned on each image for high computational efficiency. In specific, a cascade of Transformers are trained using increasing number of tokens. At test time, these models are sequentially activated starting with less tokens. Once a prediction with sufficient confidence has been produced, the inference procedure will be terminated immediately. As a consequence, the computation is unevenly allocated among "easy" and "hard" samples by adjusting the token number, yielding a considerable improvement in efficiency. Importantly, we further develop *feature-wise* and *relationship-wise* reuse mechanisms to reduce redundant computations. The former allows the downstream models to be trained on the basis of previously extracted deep features, while the later enables leveraging existing upstream self-attention relationships to learn more accurate attention maps. Illustrative examples of our method are given in Figure 1.

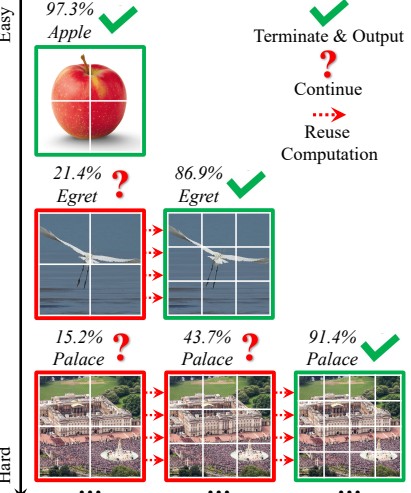

Figure 1: Examples for DVT.

Notably, DVT is designed as a general framework. Most of the state-of-the-art image recognition Transformers, such as ViT [11], DeiT [38], and T2T-ViT [55], can be straightforwardly deployed as its backbones for higher efficiency. Our method is also appealing in its flexibility. The computational cost of DVT is able to be adjusted online by simply adapting the early-termination criterion. This characteristic makes DVT suitable for the cases where the available computational resources fluctuate dynamically or a minimal power consumption is required to achieve a given performance. Both situations are ubiquitous in real-world applications (e.g., searching engines and mobile apps).

The performance of DVT is evaluated on ImageNet [9] and CIFAR [26] with T2T-ViT [55] and DeiT [38]. Experimental results show that DVT significantly improves the efficiency of the backbones. For examples, DVT reduces the computational cost of T2T-ViT by 1.6-3.6x without sacrificing accuracy. The real inference speed on a NVIDIA 2080Ti GPU is consistent with our theoretical results.

## 2 Related Work

**Vision Transformers.** Inspired by the success of Transformers on NLP tasks [10, 40, 3, 43], vision Transformers (ViT) have recently been developed for image recognition [11]. Although ViT by itself is not comparable with state-of-the-art convolutional networks (CNN) on the standard ImageNet benchmark, it attains excellent results when pre-trained on the larger JFT-300M dataset. DeiT [38] studies the training strategy of ViT and proposes a knowledge distilling-based approach, surpassing the performance of ResNet [19]. Some following works such as T2T-ViT [55], TNT [17], CaiT [39], DeepViT [62], CPVT [6], LocalViT [28] and CrossViT [5] focus on improving the architecture design of ViT. Another line of research proposes to integrate the inductive bias of CNN into Transformers [49, 8, 54, 16]. There are also attempts to adapt ViT for other vision tasks (e.g., object detection, semantic segmentation, etc.) [30, 44, 12, 20, 57, 60, 14]. The most majority of these concurrent

works represent each image with a fix number of tokens. To the best of our knowledge, we are the first to consider configuring token numbers conditioned on the inputs.

**Efficient deep networks.** Computational efficiency plays a critical role in real-world scenarios, where the executed computation translates into power consumption, carbon emission or latency. A number of works have been done on reducing the computational cost of CNNs [22, 33, 21, 53, 59, 31, 35]. However, designing efficient vision Transformers is still an under-explored topic. T2T-ViT [55] proposes a light-weighted tokens-to-token module and obtains a competitive accuracy-parameter trade-off compared to MobileNetV2 [33]. LeViT [16] accelerates the inference of Transformer models by involving convolutional layers. Swin Transformer [30] introduces an efficient shifted window-based approach in multi-stage vision Transformers. Compared to these models with fixed computational graphs, the proposed DVT framework improves the efficiency by adaptively changing the architecture of the network on a per-sample basis.

**Dynamic models.** Designing dynamic architectures is an effective approach for efficient deep learning [18]. In the context of recognition tasks, MSDNet and its variants [23, 52, 27] develop a multi-classifier CNN architecture to perform early exiting for easy samples. Another type of dynamic CNNs skips redundant layers [41, 45, 51] or channels [29] conditioned on the inputs. Besides, the spatial adaptive paradigm [15, 4, 47, 42, 46] has been proposed for efficient image and video recognition. Although these works are related to DVT on the spirit of adaptive computation, they are developed based on CNN, while DVT is tailored for vision Transformers.

Notably, our work may seem similar to the multi-exit networks [23, 52, 27, 47] from the lens of early-exiting. However, DVT differentiates itself from these existing works in several important aspects. For example, a major efficiency gain in both MSDNet [23] and RANet [52] comes from the adaptive depth, namely processing "easier" samples using fewer layers within a single network. In comparison, DVT cascades multiple Transformers to infer several entire networks at test time. Besides, DVT does not leverage the multi-scale architecture or dense connection in MSDNet [23]. Compared with RANet [52], DVT introduces the adaptive token number and the relationship reuse mechanism tailored for ViT. In addition, IMTA [27] studies the training techniques of multi-exit models, which are actually complementary to DVT. Another difference is that all these networks [23, 52, 27, 47] adopt pure convolutional models.

## 3   Dynamic Vision Transformer

Vision Transformers [11, 17, 38, 55] split each 2D image into 1D tokens, while model their long range interaction with the self-attention mechanism [40]. As aforementioned, to correctly recognize some "hard" images and achieve high accuracy, the number of tokens usually needs to be large, leading to the quadratically grown computational cost. However, "easier" images that make up the bulk of the datasets typically require far fewer tokens and much less costs (as shown in Table 1). Inspired by this observation, we propose a *Dynamic Vision Transformers* (DVT), aiming to improve the computational efficiency of Transformers via adaptively reducing the number of representative tokens for each input.

In specific, we propose to deploy multiple Transformers trained with increasing number of tokens, such that one can sequentially activate them for each test image until obtaining a convincing prediction (e.g., with sufficient confidence). The computation is allocated unevenly across different samples for improving the overall efficiency. It is worth noting that, if all the Transformers are learned separately, the computation performed by upstream models will simply be abandoned once a downstream Transformer is activated, resulting in considerable inefficiency. To alleviate this problem, we introduce the efficient feature and relationship reuse mechanisms.

### 3.1   Overview

**Inference.** We start by describing the inference procedure of DVT, which is shown in Figure 2. For each test sample, we first coarsely represent it using a small number of 1D token embeddings. This can be achieved by either straightforwardly flattening the split image patches [11, 17] or leveraging techniques like the tokens-to-token module [55]. We infer a vision Transformer with these few tokens to obtain a quick prediction. This process enjoys high efficiency since the computational cost of Transformers grows quadratically with respect to token number. Then the prediction will be evaluated with certain criterion to determine whether it is reliable enough to be retrieved immediately. In this paper, early-termination is performed when the model is sufficiently confident (details in Section 3.3).

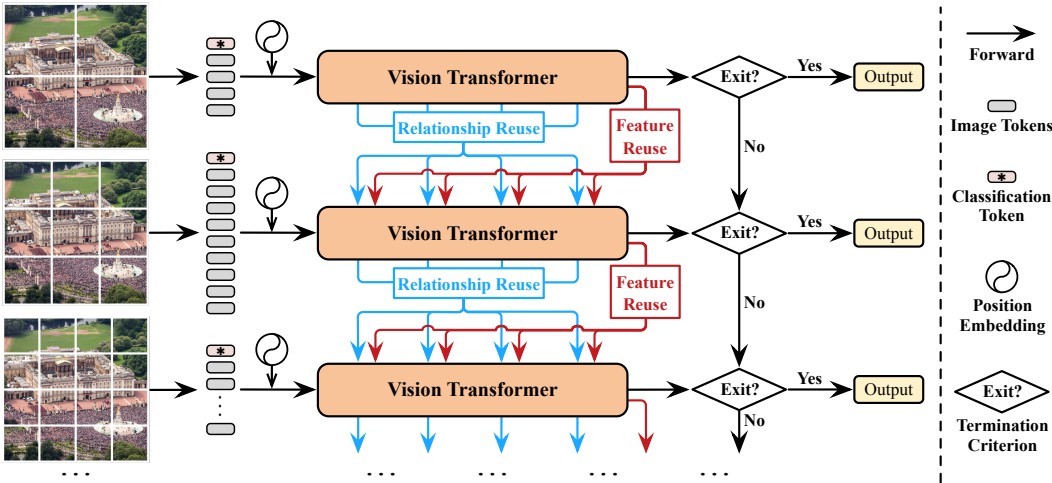

Figure 2: An overview of *Dynamic Vision Transformers* (DVT). Under the objective of configuring proper token numbers conditioned on the inputs, we cascade multiple Transformers with increasing number of tokens. At test time, they are sequentially activated until a convincing prediction (e.g. sufficiently confident) has been obtained or the final model has been inferred. The feature and relationship reuse mechanisms allow reusing computation across different Transformers.

Once the prediction fails to meet the termination criterion, the original input image will be split into more tokens for more accurate but computationally more expensive inference. Note that, here the dimension of each token embedding remains unchanged, while the number of tokens increases, enabling more fine-grained representation. An additional Transformer with the same architecture as the previous one but different parameters will be activated. By design, this stage trades off computation for higher accuracy on some "difficult" test samples. To improve the efficiency, the new model can reuse the previously learned features and relationships, which will be introduced in Section 3.2. Similarly, after obtaining a new prediction, the termination criterion will be applied, and the above procedure will proceed until the sample exits or the final Transformer has been inferred.

**Training.** For training, we simply train DVT to produce correct predictions at all exits (i.e., each with the corresponding number of tokens). Formally, the optimization objective is

$$\text{minimize} \quad \frac{1}{|\mathcal{D}_{\text{train}}|} \sum_{(\boldsymbol{x},y) \in \mathcal{D}_{\text{train}}} \left[ \sum_i L_{\text{CE}}(\boldsymbol{p}_i, y) \right], \tag{1}$$

where $(\boldsymbol{x}, y)$ denote a sample in the training set $\mathcal{D}_{\text{train}}$ and its corresponding label. We adopt the standard cross-entropy loss function $L_{\text{CE}}(\cdot)$, while $\boldsymbol{p}_i$ denotes the softmax prediction probability output by the $i^{\text{th}}$ exit. We find that such a simple training objective works well in practice.

**Transformer backbone.** DVT is proposed as a general and flexible framework. It can be built on top of most existing vision Transformers like ViT [11], DeiT [38] and T2T-ViT [55] to improve their efficiency. The architecture of Transformers simply follows the implementation of these backbones.

## 3.2 Feature and Relationship Reuse

An important challenge to develop our DVT approach is how to facilitate the *reuse* of computation. That is, once a downstream Transformer with more tokens is inferred, it is obviously inefficient if the computation performed in previous models is abandoned. The upstream models, although being based on smaller number of input tokens, are trained with the same objective, and have extracted valuable information for fulfilling the task. Therefore, we propose two mechanisms to reuse the learned deep features and self-attention relationships. Both of them are able to improve the test accuracy significantly by involving minimal extra computational cost.

**Background.** For the ease of introduction, we first revisit the basic formulation of vision Transformers. The Transformer encoders consist of alternatively stacked multi-head self-attention (MSA) and multi-layer perceptron (MLP) blocks [40, 11]. The layer normalization (LN) [2] and residual

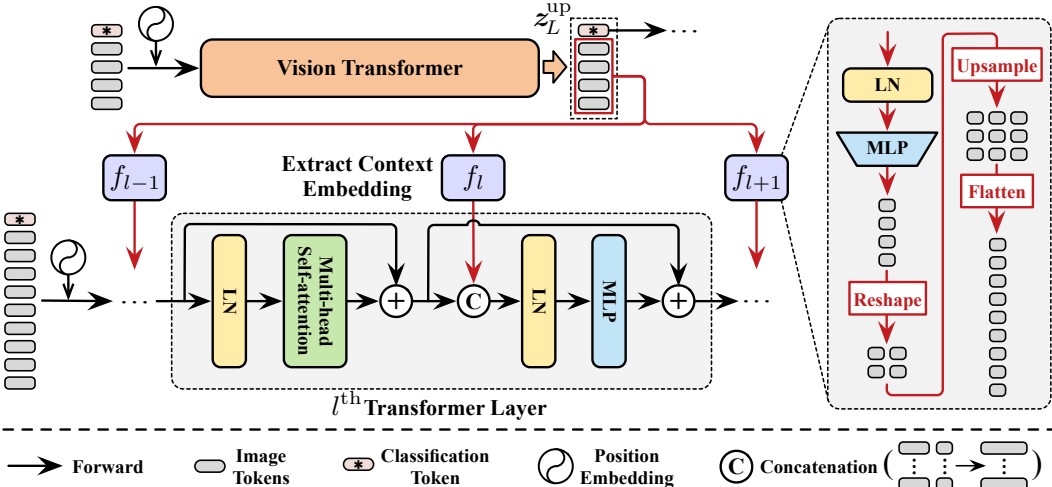

Figure 3: Illustration of the feature reuse mechanism. A layer-wise context embedding is learned based on the final representations output by the upstream model, i.e., $z_L^{\text{up}}$, and integrated into the MLP block of each downstream Transformer layer.

connection [19] are applied before and after each block, respectively. Let $z_l \in \mathbb{R}^{N \times D}$ denote the output of the $l^{\text{th}}$ Transformer layer, where $N$ is the number of tokens for each sample, and $D$ is the dimension of each token. Note that $N = HW + 1$, which corresponds to $H \times W$ patches of the original image and a single learnable classification token. Formally, we have

$$z_l' = \text{MSA}(\text{LN}(z_{l-1})) + z_{l-1}, \qquad l \in \{1, \ldots, L\}, \tag{2}$$

$$z_l = \text{MLP}(\text{LN}(z_l')) + z_l', \qquad l \in \{1, \ldots, L\}, \tag{3}$$

where $L$ is the total number of layers in the Transformer. The classification token in $z_L$ will be fed into a LN layer followed by a fully-connected layer for the final prediction. For simplicity, here we omit the details on the position embedding, which is unrelated to our main idea. No modification is performed on it in addition to the configurations of backbones.

**Feature reuse.** All the Transformers in DVT share the same goal of extracting discriminative representations for accurate recognition. Therefore, it is straightforward that downstream models should be learned on the basis of previously obtained deep features, rather than extracting features from scratch. The former is more efficient since the computation performed in an upstream model contributes to both itself and the successive models. To implement this idea, we propose a feature reuse mechanism (see: Figure 3). In specific, we leverage the image tokens output by the final layer of the upstream Transformer, i.e., $z_L^{\text{up}}$, to learn a layer-wise embedding $\mathbf{E}_l$ for the downstream model:

$$\mathbf{E}_l = f_l(z_L^{\text{up}}) \in \mathbb{R}^{N \times D'}. \tag{4}$$

Herein, $f_l \colon \mathbb{R}^{N \times D} \to \mathbb{R}^{N \times D'}$ consists of a sequence of operations starting with a LN-MLP ($\mathbb{R}^D \to \mathbb{R}^{D'}$), which introduces nonlinearity and allows more flexible transformations. Then the image tokens are reshaped to the corresponding locations in the original image, upsampled and flattened to match the token number of the downstream model. Typically, we use a small $D'$ for an efficient $f_l$.

Consequently, the embedding $\mathbf{E}_l$ is injected into the downstream model, providing prior knowledge on recognizing the input image. Formally, we replace Eq. (3) by:

$$z_l = \text{MLP}(\text{LN}(\text{Concat}(z_l', \mathbf{E}_l))) + z_l', \qquad l \in \{1, \ldots, L\}, \tag{5}$$

where $\mathbf{E}_l$ is concatenated with the intermediate tokens $z_l'$. We simply increase the dimension of LN and the first layer of MLP from $D$ to $D+D'$. Since $\mathbf{E}_l$ is based on the upstream outputs $z_L^{\text{up}}$ that have less tokens than $z_l'$, it actually concludes the context information of the input image for each token in $z_l'$. Therefore, we name $\mathbf{E}_l$ as the *context embedding*. Besides, we do not reuse the classification token and pad zero for it in Eq. (5), which we empirically find beneficial for the performance. Intuitively, Eqs. (4) and (5) allow training the downstream model to flexibly exploit the information within $z_L^{\text{up}}$ on a per-layer basis, under the objective of minimizing the final recognition loss (Eq. (1)). This feature reuse formulation can also be interpreted as implicitly enlarging the depth of the model.

**Relationship reuse.** A prominent advantage of vision Transformers is that their self-attention blocks enable integrating information across the entire image, which effectively models the long-range dependencies in the data. Typically, the models need to learn a group of attention maps at each layer to describe the relationships among tokens. Apart from the deep features mentioned above, the downstream models also have access to the self-attention maps produced in previous models. We argue that these learned relationships are also capable of being reused to facilitate the learning of downstream Transformers.

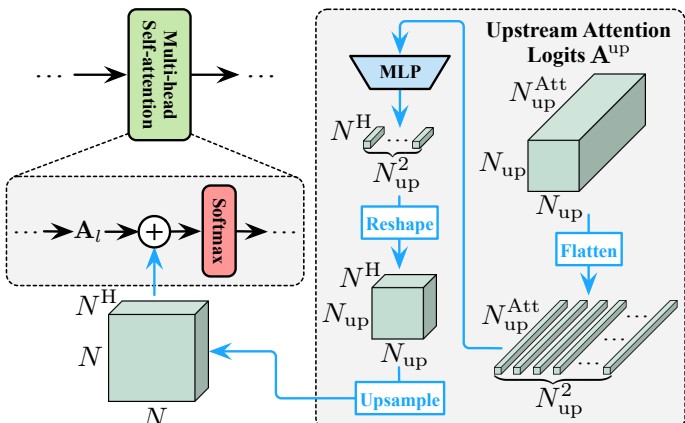

Figure 4: Illustration of the relationship reuse mechanism. We leverage the learned self-attention relationships from all upstream layers and attention heads, i.e., $\mathbf{A}^{\mathrm{up}}$, to refine the downstream attention maps. The addition operation of logits is adopted. Note that $N^{\mathrm{H}}$ denotes the head number for multi-head self-attention.

Given the input representation $z_l$, the self-attention is performed as follows. First, the query, key and value matrices $\mathbf{Q}_l$, $\mathbf{K}_l$ and $\mathbf{V}_l$ are computed via linear projections:

$$\mathbf{Q}_l = z_l \mathbf{W}_l^{\mathrm{Q}}, \quad \mathbf{K}_l = z_l \mathbf{W}_l^{\mathrm{K}}, \quad \mathbf{V}_l = z_l \mathbf{W}_l^{\mathrm{V}}, \tag{6}$$

where $\mathbf{W}_l^{\mathrm{Q}}$, $\mathbf{W}_l^{\mathrm{K}}$ and $\mathbf{W}_l^{\mathrm{V}}$ are weight matrices. Then the attention map is calculated by a scaled dot-product operation with softmax to aggregate the values of all tokens, namely

$$\mathrm{Attention}(z_l) = \mathrm{Softmax}(\mathbf{A}_l)\mathbf{V}_l, \quad \mathbf{A}_l = \mathbf{Q}_l\mathbf{K}_l^\top/\sqrt{d}. \tag{7}$$

Here $d$ is the hidden dimension of $\mathbf{Q}$ or $\mathbf{K}$, and $\mathbf{A}_l \in \mathbb{R}^{N \times N}$ denotes the logits of the attention map. Note that we omit the details on the multi-head attention mechanism for clarity, where $\mathbf{A}_l$ may include multiple attention maps. Such a simplification does not affect the description of our method.

For relationship reuse, we first concatenate the attention logits produced by all layers of the upstream model (i.e., $\mathbf{A}_l^{\mathrm{up}}, l \in \{1, \ldots, L\}$):

$$\mathbf{A}^{\mathrm{up}} = \mathrm{Concat}(\mathbf{A}_1^{\mathrm{up}}, \mathbf{A}_2^{\mathrm{up}}, \ldots, \mathbf{A}_L^{\mathrm{up}}) \in \mathbb{R}^{N_{\mathrm{up}} \times N_{\mathrm{up}} \times N_{\mathrm{up}}^{\mathrm{Att}}}, \tag{8}$$

where $N_{\mathrm{up}}$ and $N_{\mathrm{up}}^{\mathrm{Att}}$ denote the number of tokens and all attention maps in the upstream model, respectively. Typically, we have $N_{\mathrm{up}}^{\mathrm{Att}} = N^{\mathrm{H}}L$, where $N^{\mathrm{H}}$ is the number of heads for the multi-head attention and $L$ is the number of layers. Then the downstream Transformer learns attention maps by leveraging both its own tokens and $\mathbf{A}^{\mathrm{up}}$ simultaneously. Formally, we replace Eq. (7) by

$$\mathrm{Attention}(z_l) = \mathrm{Softmax}(\mathbf{A}_l + r_l(\mathbf{A}^{\mathrm{up}}))\mathbf{V}_l, \quad \mathbf{A}_l = \mathbf{Q}_l\mathbf{K}_l^\top/\sqrt{d}, \tag{9}$$

where $r_l(\cdot)$ is a transformation network that integrates the information provided by $\mathbf{A}^{\mathrm{up}}$ to refine the downstream attention logits $\mathbf{A}_l$. The architecture of $r_l(\cdot)$ is presented in Figure 4, which includes a MLP for nonlinearity followed by an upsample operation to match the size of attention maps. For multi-head attention, the output dimension of the MLP will be set to the number of heads.

Notably, Eq. (9) is a simple but flexible formulation. For one thing, each self-attention block in the downstream model has access to all the upstream attention heads in both shallow and deep layers, and hence can be trained to leverage multi-level relationship information on its own basis. For another, as the newly generated attention maps and the reused relationships are combined in logits, their relative importance can be automatically learned by adjusting the magnitude of logits. It is also worth noting that the regular upsample operation cannot be directly applied in $r_l(\cdot)$. To illustrate this issue, we take

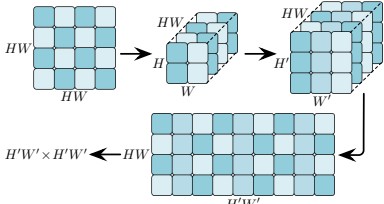

Figure 5: An example for the upsample operation in $r_l(\cdot)$.

upsampling a $HW \times HW$ ($H\!=\!W\!=\!2$) attention map to $H'W' \times H'W'$ ($H'\!=\!W'\!=\!3$) for example in Figure 5. Since each of its rows and columns corresponds to $H \times W$ image tokens, we reshape the rows or columns back to $H \times W$, scale them to $H' \times W'$, and then flatten them to $H'W'$ vectors.

### 3.3 Adaptive Inference

As aforementioned, the proposed DVT framework progressively increases the number of tokens for each test sample and performs early-termination, such that "easy" and "hard" images can be processed using varying tokens with uneven computational cost, improving the overall efficiency. Specifically, at the $i^{\text{th}}$ exit that produces the softmax prediction $\boldsymbol{p}_i$, the largest entry of $\boldsymbol{p}_i$, i.e., $\max_j p_{ij}$ (defined as confidence [23, 52, 47]), is compared with a threshold $\eta_i$. If $\max_j p_{ij} \geq \eta_i$, the inference will stop by adopting $\boldsymbol{p}_i$ as the output. Otherwise, the image will be represented using more tokens to activate the downstream Transformer. We always adopt a zero-threshold for the final Transformer.

The values of $\{\eta_1, \eta_2, \ldots\}$ are solved on the validation set. We assume a *budgeted batch classification* [23] setting, where DVT needs to recognize a set of samples $\mathcal{D}_{\text{val}}$ within a given computational budget $B > 0$. Let $\text{Acc}(\mathcal{D}_{\text{val}}, \{\eta_1, \eta_2, \ldots\})$ and $\text{FLOPs}(\mathcal{D}_{\text{val}}, \{\eta_1, \eta_2, \ldots\})$ denote the accuracy and computational cost on $\mathcal{D}_{\text{val}}$ when using the thresholds $\{\eta_1, \eta_2, \ldots\}$. The optimal thresholds can be obtained by solving the following optimization problem:

$$\underset{\eta_1, \eta_2, \ldots}{\text{maximize}} \quad \text{Acc}(\mathcal{D}_{\text{val}}, \{\eta_1, \eta_2, \ldots\}) \quad \text{subject to} \quad \text{FLOPs}(\mathcal{D}_{\text{val}}, \{\eta_1, \eta_2, \ldots\}) \leq B. \quad (10)$$

Due to the non-differentiability, we solve this problem with the genetic algorithm [48] in this paper.

## 4 Experiments

In this section, we empirically validate the proposed DVT on ImageNet [9] and CIFAR-10/100 [26]. Ablation studies and visualization are presented on ImageNet to give a deeper understanding of our method. Code and pre-trained models based on PyTorch are available at `https://github.com/blackfeather-wang/Dynamic-Vision-Transformer`. We also provide the implementation using the MindSpore framework and the models trained on a cluster of Ascend AI processors at `https://github.com/blackfeather-wang/Dynamic-Vision-Transformer-MindSpore`.

**Datasets.** (1) ImageNet is a 1,000-class dataset from ILSVRC2012 [9], containing 1.2 million images for training and 50,000 images for validation. (2) CIFAR-10/100 datasets [26] contain 32x32 colored images in 10/100 classes. Both of them consist of 50,000 images for training and 10,000 images for testing. For all the three datasets, we adopt the same data pre-processing and data augmentation policy as [19, 24, 23]. In addition, we solve the confidence thresholds stated in Section 3.3 on the training set, which we find achieves similar performance to adopting cross-validation.

**Backbones.** Our experiments are based on several state-of-the-art vision Transformers, namely T2T-ViT-12 [55], T2T-ViT-14 [55], and DeiT-small (w/o distillation) [38]. Unless otherwise specified, we deploy DVT with three exits, corresponding to representing the images as 7x7, 10x10 and 14x14 tokens[3]. For fair comparisons, our implementation exploits the official code of the backbones, and adopts exactly the same training hyper-parameters. More training details can be found in Appendix A. The number of FLOPs is calculated using the *fvcore* toolkit provided by Facebook AI Research, which is also used in Detectron2 [50], PySlowFast [13], and ClassyVision [1].

**Implementation details.** For feature reuse, the hidden size and output size of the MLP in $f_l(\cdot)$ are set to 128 and 48. In relationship reuse, for implementation efficiency, we share the same hidden state across the MLPs of all $r_l(\cdot)$, such that $r_l(\mathbf{A}^{\text{up}}), l \in \{1, \ldots, L\}$ can be obtained at one time in concatenation by implementing a single large MLP, whose hidden size and output size are $3N^{\text{H}}L$ and $N^{\text{H}}L$. Note that $N^{\text{H}}$ is the head number of multi-head attention and $L$ is the layer number.

### 4.1 Main Results

**Results on ImageNet** are shown in Figures 6 and 7, where T2T-ViT [55] and DeiT [38] are implemented as backbones respectively. As stated in Section 3.3, we vary the average computational budget, solve the confidence thresholds, and evaluate the corresponding validation accuracy. The performance of DVT is plotted in gray curves, with the best accuracy under each budget plotted in black curves. We also compare our method with several highly competitive baselines, i.e., TNT [17], LocalViT [28], CrossViT [5], PVT [44], ViT [11] and ResNet [19]. It can be observed that DVT consistently reduces the computational cost of the backbones. For example, DVT achieves the 82.3% accuracy with 3.6x less FLOPs compared with the vanilla T2T-ViT. When the budget ranges among 0.5-2 GFLOPs, DVT has ∼1.7-1.9x less computation than T2T-ViT with the same performance. Notably, our method can flexibly attain all the points on each curve by simply adjusting the values of confidence thresholds with a single DVT.

---

[3]Although 4x4 tokens are also used as an example in Section 1, we find starting with 7x7 is more efficient.

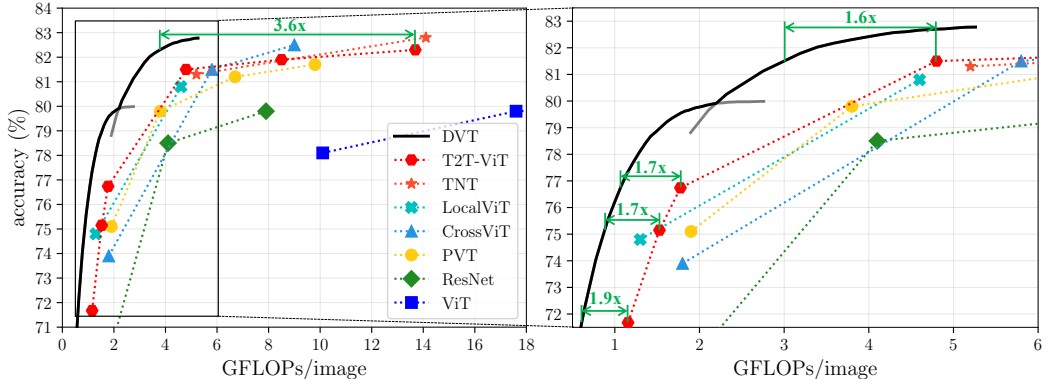

Figure 6: Top-1 accuracy v.s. GFLOPs on ImageNet. DVT is implemented on top of T2T-ViT-12/14.

Table 2: The practical speed of DVT.

| Models | ImageNet (NVIDIA 2080Ti, bs=128) | |
| | Top-1 Acc. | Throughput |
|---|---|---|
| T2T-ViT-7 | 71.68% | 1574 img/s |
| DVT | **78.48%** (↑6.80%) | 1574 img/s |
| T2T-ViT-10 | 75.15% | 1286 img/s |
| DVT | **79.74%** (↑4.59%) | 1286 img/s |
| T2T-ViT-12 | 76.74% | 1121 img/s |
| DVT | **80.43%** (↑3.69%) | 1128 img/s |
| T2T-ViT-14 | 81.50% | 619 img/s |
| DVT | 81.50% | **877 img/s** (↑1.42x) |
| T2T-ViT-19 | 81.93% | 382 img/s |
| DVT | 81.93% | **666 img/s** (↑1.74x) |

Table 3: Performance of DVT on CIFAR-10/100.

| Models | CIFAR-10 | | CIFAR-100 | |
| | Top-1 Acc. | GFLOPs | Top-1 Acc. | GFLOPs |
|---|---|---|---|---|
| T2T-ViT-10 | 97.21% | 1.53 | 85.44% | 1.53 |
| DVT | 97.21% | **0.50** (↓3.1x) | 85.45% | **0.54** (↓2.8x) |
| T2T-ViT-12 | 97.45% | 1.78 | 86.23% | 1.78 |
| DVT | 97.46% | **0.52** (↓3.4x) | 86.26% | **0.61** (↓2.9x) |
| T2T-ViT-14 | 98.19% | 4.80 | 89.10% | 4.80 |
| DVT | 98.19% | **0.77** (↓6.2x) | 89.11% | **1.62** (↓3.0x) |
| T2T-ViT-19 | 98.43% | 8.50 | 89.37% | 8.50 |
| DVT | 98.43% | **1.44** (↓5.9x) | 89.38% | **1.74** (↓4.9x) |
| T2T-ViT-24 | 98.53% | 13.69 | 89.62% | 13.69 |
| DVT | 98.53% | **1.49** (↓9.2x) | 89.63% | **1.86** (↓7.4x) |

**Practical efficiency of DVT**. We test the actual speed of DVT on a NVIDIA 2080Ti GPU under a batch inference setting, where a mini-batch of data is fed into the model at a time. After inferring each Transformer, the samples that meet the early-termination criterion will exit, with the remaining images fed into the downstream Transformer. The results are presented in Table 2. Here we adopt a two-exit DVT based on T2T-ViT-12 using 7x7 and 14x14 tokens, which we find more efficient in practice. All other implementation details remain unchanged. One can observe that DVT improves the accuracy of small models (T2T-ViT-7/10/12) by 3.7-6.8% with the same inference speed, while accelerates the inference of the large T2T-ViT-14/19 models by 1.4-1.7x without sacrificing performance.

**Results on CIFAR** are presented in Table 3. Following the common practice [11, 55, 17, 38], we resize the CIFAR images to 224x224, and fine-tune the T2T-ViT and DVT models in Figure 6. The official code and training configurations provided by [55] are utilized. We report the computational cost of DVT when it achieves the competitive performance with baselines. Our proposed method is shown to consume ∼3-9x less computation compared with T2T-ViT.

Table 4: Comparisons between DVT and three state-of-the-art methods for improving the computational efficiency of vision Transformers, i.e., Data distillation [38], DynamicViT [32] and Patch slimming [36]. Both DVT and the baselines are implemented on top of DeiT-small. Since the computational cost of DVT can be adjusted online, we match the FLOPs or the accuracy of DVT with the baselines, respectively, to see the difference.

| | DeiT-small | Data distillation | DVT | | DynamicViT | DVT | | Patch slimming | DVT | |
|---|---|---|---|---|---|---|---|---|---|---|
| Top-1 Acc. | 79.80% | 81.20% | 81.20% | **81.67%** | 79.30% | 79.30% | **80.40%** | 79.40% | 79.40% | **79.81%** |
| GFLOPs/image | 4.61 | 4.63 | **3.61** | 4.63 | 2.90 | **2.37** | 2.90 | 2.60 | **2.41** | 2.60 |

**Comparisons with SOTA baselines.** In Table 4, we compare DVT with several recently proposed approaches for facilitating efficient vision Transformers on ImageNet. Both knowledge distillation (KD) [38] and patch pruning based methods [32, 36] are considered. One can observe that DVT outperforms the baselines with similar FLOPs, or has significantly lower computational cost with the same accuracy. Besides, one can expect that DVT is compatible with these techniques, i.e., DVT can also be trained with KD to further boost the accuracy or leverage patch pruning to reduce the FLOPs.

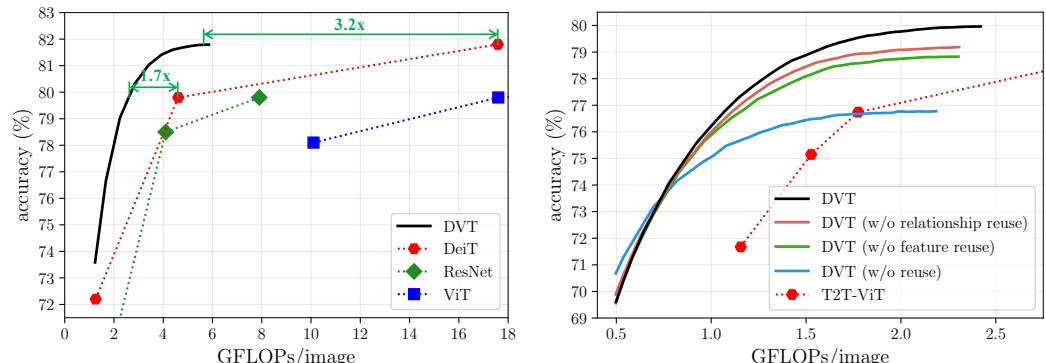

Figure 7: Performance of DeiT-based DVT on ImageNet. DeiT-small is used as the backbone.

Figure 8: Performance of the DVT based on T2T-ViT-12 with and without the reuse mechanisms.

**Comparisons with existing early-exiting networks** are presented in Table 5. We adopt the same adaptive strategy (see: Section 3.3) for all the baselines. The Top-1 accuracy on ImageNet under different computational budgets is reported.

Table 5: DVT v.s. existing multi-exit models with the same adaptive inference strategy. The accuracy under each budget is reported.

| Networks | Top-1 Acc. | | | | | |
|---|---|---|---|---|---|---|
| | 0.75G | 1.00G | 1.25G | 1.50G | 1.75G | 2.00G |
| MSDNet [23] | 69.82% | 71.24% | 72.73% | 73.66% | 73.99% | 74.20% |
| IMTA-MSDNet [27] | 70.84% | 71.92% | 73.41% | 74.31% | 74.64% | 74.94% |
| RANet [52] | 70.48% | 72.24% | 73.57% | 74.56% | 75.02% | 75.10% |
| DVT (T2T-ViT-12) | **73.70%** | **76.22%** | **77.89%** | **78.89%** | **79.47%** | **79.75%** |

One can observe that DVT significantly outperforms these baselines.

### 4.2 Ablation Study

**Effectiveness of feature and relationship reuse.** We conduct experiments by ablating one or both of the reuse mechanisms. For a clear comparison, we first deactivate the early-termination, and report the accuracy and GFLOPs corresponding to each exit in Table 6. The three-exit DVT based on T2T-ViT-

Table 6: Effects of feature (F) and relationship (R) reuse. The percentages in brackets denote the additional computation compared to baselines involved by the reuse mechanisms.

| Reuse | | 1$^{st}$ Exit (7x7) | | 2$^{nd}$ Exit (10x10) | | 3$^{rd}$ Exit (14x14) | |
|---|---|---|---|---|---|---|---|
| F | R | Top-1 Acc. | GFLOPs | Top-1 Acc. | GFLOPs | Top-1 Acc. | GFLOPs |
| | | **70.33%** | 0.47 | 73.54% | 1.37 | 76.74% | 3.15 |
| ✓ | | 69.42% | 0.47 | 75.31% | 1.43$_{(4.4\%)}$ | 79.21% | 3.31$_{(5.1\%)}$ |
| | ✓ | 69.03% | 0.47 | 75.34% | 1.41$_{(2.9\%)}$ | 78.86% | 3.34$_{(6.0\%)}$ |
| ✓ | ✓ | 69.04% | 0.47 | **75.65%** | 1.46$_{(6.6\%)}$ | **80.00%** | 3.50$_{(11.1\%)}$ |

12 is considered. One can observe that both the two reuse mechanisms are able to significantly boost the accuracy of DVT at the 2$^{nd}$ and 3$^{rd}$ exits with at most 6% additional computation, while they are compatible with each other to further improve the performance. We also find that involving computation reusing slightly hurts the accuracy at the 1$^{st}$ exit, which may be attributed to the compromise made by the first Transformer for downstream models. However, once the early-termination is adopted, this difference only results in trivial disadvantage when the computational budget is very small, as shown in Figure 8. DVT outperforms the baseline significantly in most cases.

**Design choices for the reuse mechanisms.** Here we study the design of the feature and relationship reuse mechanisms. For experimental efficiency, we consider a two-exit DVT based on T2T-ViT-12 using 7x7 and 10x10 tokens, while enlarge the batch size and the initial learning rate by 4 times. Such a training setting slightly degrades the accuracy of DVT, but it is still reliable to reflect the difference between different design variants. We deactivate early-termination, and report the performance of each exit. Notably, as the FLOPs of 1$^{st}$ exit remain unchanged (i.e., 0.47G), we do not present it.

We consider four variants of feature reuse in Table 7: (1) reusing features from the corresponding upstream layer instead of the final layer; (2) reusing classification token; (3) only performing feature reuse in the first layer of the downstream model; (4) removing the LN in $f_l(\cdot)$. One can see that taking final tokens of the upstream model and reusing them in each downstream layer are both important.

Ablation results for relationship reuse are presented in Table 8. We consider four variants as well: (1) only reusing the attention logits from the corresponding upstream layer; (2) only reusing the attention logits from the final upstream layer; (3) replacing the MLP by a linear layer in $r_l(\cdot)$; (4) adopting naive upsample operation instead of what is shown in Figure 5; The results indicate that it is beneficial to enable each downstream layer to flexibly reuse all upstream attention logits. Besides, naive upsampling significantly hurts the performance.

Table 7: Ablation studies for feature reuse.

| Ablation | 1st Exit (7x7) Top-1 Acc. | 2nd Exit (10x10) Top-1 Acc. | GFLOPs |
|---|---|---|---|
| w/o reuse | **70.08%** | 73.61% | 1.37 |
| Layer-wise feature reuse | 69.84% | 74.31% | 1.43 |
| Reuse classification token | 69.79% | 74.70% | 1.43 |
| Remove $f_l(\cdot), l \geq 2$ | 69.33% | 74.73% | 1.38 |
| Remove LN in $f_l(\cdot)$ | 69.63% | 75.05% | 1.42 |
| Ours | 69.44% | **75.23%** | 1.43 |

Table 8: Ablation studies for relationship reuse.

| Ablation | 1st Exit (7x7) Top-1 Acc. | 2nd Exit (10x10) Top-1 Acc. | GFLOPs |
|---|---|---|---|
| w/o reuse | **70.08%** | 73.61% | 1.37 |
| Layer-wise relationship reuse | 69.63% | 73.89% | 1.38 |
| Reuse final-layer relationships | 69.25% | 74.31% | 1.39 |
| MLP→Linear | 69.20% | 73.84% | 1.38 |
| Naive upsample | 69.60% | 73.34% | 1.41 |
| Ours | 69.50% | **74.91%** | 1.41 |

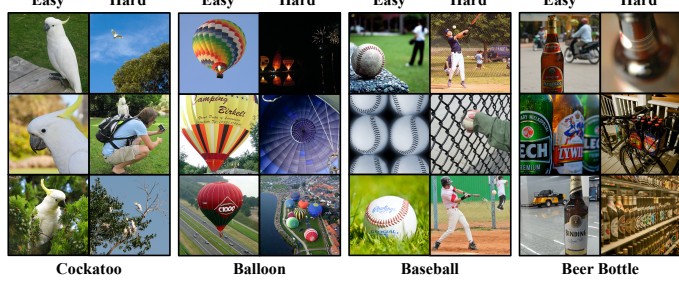

Figure 9: Visualization of the "easy" and "hard" samples in DVT.

Figure 10: Numbers of images exiting at different exits with varying computational budgets.

**Early-termination criterion.** We vary the criterion for adaptive inference based on DVT (T2T-ViT-12) and report the accuracy under several computational budgets in Table 9. Two variants are considered: (1) adopting the entropy of softmax prediction to determine whether to exit [37]; (2) performing random exiting with the same exit proportion as DVT. The simple but effective confidence-based criterion achieves better performance than both of them.

Table 9: Comparisons of early-termination criterions. The accuracy under each budget is reported.

| Ablation | Top-1 Acc. | | | |
|---|---|---|---|---|
| | 0.75G | 1.00G | 1.25G | 1.50G |
| Randomly Exit | 70.19% | 71.66% | 72.61% | 73.59% |
| Entropy-based | 73.41% | 75.21% | 77.08% | 78.40% |
| Confidence-based (ours) | **73.70%** | **76.22%** | **77.89%** | **78.89%** |

**Vision Transformers with adaptive depth.** We test applying the adaptive depth paradigm [23, 52] to ViT in Table 10. Two classifiers are uniformly added to the intermediate layers of T2T-ViT-14, and the early-exiting is performed

Table 10: Effects of applying the adaptive depth mechanism to vision Transformers. The accuracy under each budget is reported.

| Ablation | Top-1 Acc. | | | | |
|---|---|---|---|---|---|
| | 2.00G | 2.50G | 3.00G | 3.50G | 4.00G |
| T2T-ViT-14 with adaptive depth | 76.09% | 78.88% | 79.46% | 79.54% | 79.55% |
| DVT (T2T-ViT-14) | **79.24%** | **80.61%** | **81.47%** | **82.10%** | **82.42%** |

with the three exits in the same way as DVT. One can observe that the layer-adaptive T2T-ViT achieves significantly lower accuracy than DVT using identical computational budgets. This inferior performance may be alleviated by improving the network architecture, as shown in [23].

### 4.3 Visualization

Figure 9 shows the images that are first correctly classified at the 1st and 3rd exits of the DVT (T2T-ViT-12). The former are recognized as "easy" samples, while the later are considered to be "hard". One can observe that "easy" samples usually depict the recognition objectives in clear and canonical poses and sufficiently large resolution. On the contrary, "hard" samples may contain complex scenes and non-typical poses or only include a small part of the objects, and require a finer representation using more tokens. Figure 10 presents the numbers of images that exit at different exits when the computational budget increases. The plot shows that the accuracy of DVT is significantly improved with more images exiting later, which is achieved by changing the confidence thresholds online.

### 5 Conclusion

In this paper, we sought to optimally configure a proper number of tokens for each individual image in vision Transformers, and hence proposed the *Dynamic Vision Transformer* (DVT) framework. DVT processes each test input by sequentially activating a cascade of Transformers using increasing numbers of tokens, until an appropriate token number is reached (measured by the prediction confidence). We further introduce the feature and relationship reuse mechanisms to facilitate efficient computation reuse. Extensive experiments indicate that DVT significantly improves the computational efficiency on top of state-of-the-art vision Transformers, both theoretically and empirically.

## Acknowledgements

This work is supported in part by the National Science and Technology Major Project of the Ministry of Science and Technology of China under Grants 2018AAA0100701, the National Natural Science Foundation of China under Grants 61906106 and 62022048, and Huawei Technologies Ltd. In particular, we appreciate the valuable discussions with Chenghao Yang.

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
