# Appendix

## A  Training Details

The proposed DVT framework is implemented based on the official code of T2T-ViT[4] [55] and DeiT[5] [38] when these two models are used as backbones, respectively. The training of DVT follows exactly the same configurations as the backbones, which are also recommended by their official implementations. The details on optimizer, learning rate schedule, batch size and other hyper-parameters can be easily found in their papers or code. Note that a number of regularization and data augmentation techniques are exploited, including RandAugment [7], Random Erasing [61], Label Smoothing [34], Mixup [58], Cutmix [56], and stochastic depth [25]. We train all the models with 8 NVIDIA V100 GPUs.

Table 11: Effects when the location for performing feature reuse varies.

| Ablation | 1st Exit (7x7) Top-1 Acc. | 2nd Exit (10x10) Top-1 Acc. | GFLOPs |
|---|---|---|---|
| w/o reuse | **70.08%** | 73.61% | 1.37 |
| Reuse on shallow half of downstream layers | 69.67% | 75.02% | 1.40 |
| Reuse on deep half of downstream layers | 69.94% | 74.57% | 1.40 |
| Reuse on all downstream layers | 69.44% | **75.23%** | 1.43 |

Table 12: Effects when the location for performing relationship reuse varies.

| Ablation | 1st Exit (7x7) Top-1 Acc. | 2nd Exit (10x10) Top-1 Acc. | GFLOPs |
|---|---|---|---|
| w/o reuse | **70.08%** | 73.61% | 1.37 |
| Reuse on shallow half of downstream layers | 69.72% | 74.89% | 1.40 |
| Reuse on deep half of downstream layers | 70.00% | 73.68% | 1.40 |
| Reuse on all downstream layers | 69.50% | **74.91%** | 1.41 |

Table 13: Effects of taking attention logits from varying upstream layers.

| Ablation | 1st Exit (7x7) Top-1 Acc. | 2nd Exit (10x10) Top-1 Acc. | GFLOPs |
|---|---|---|---|
| w/o reuse | **70.08%** | 73.61% | 1.37 |
| Reuse relationships from shallow half of upstream layers | 69.93% | 74.67% | 1.40 |
| Reuse relationships from deep half of upstream layers | 69.55% | 74.58% | 1.40 |
| Reuse relationships from all upstream layers | 69.50% | **74.91%** | 1.41 |

## B  Additional Results

**Which downstream layers benefit more from reuse?** To shed light on the layer-wise reuse paradigm in the downstream model, we test performing feature or relationship reuse only in the shallow/deep half of downstream layers. The same experimental protocol as ablating the design of reuse mechanisms is adopted. The results are shown in Tables 11 and 12. Obviously, it is more important to reuse upstream features/relationships at the shallow layers. This phenomenon indicates that the main effects of the proposed reuse mechanisms lie in helping the first several transformer layers to rapidly extract discriminative representations or to learn accurate attention maps. The successive layers focus more on further improving the shallow features, while less on leveraging upstream information, which may have been effectively integrated into the shallow layers.

**Which upstream layers contribute more to relationship reuse?** In Table 13, we further test only taking the attention logits from the shallow/deep half of upstream layers in relationship reuse. One can observe that both shallow and deep relationships are important for boosting the accuracy of the

---

[4]`https://github.com/yitu-opensource/T2T-ViT`
[5]`https://github.com/facebookresearch/deit`

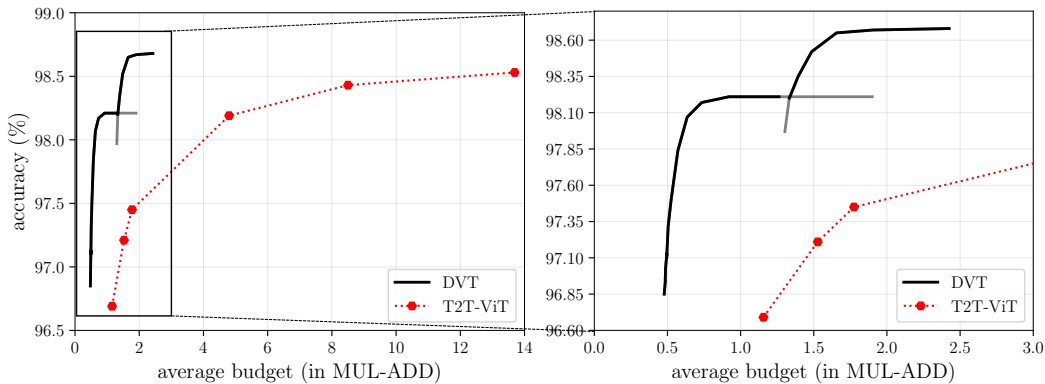

Figure 11: Top-1 accuracy v.s. GFLOPs on CIFAR-10. DVT is implemented on top of T2T-ViT-12/14.

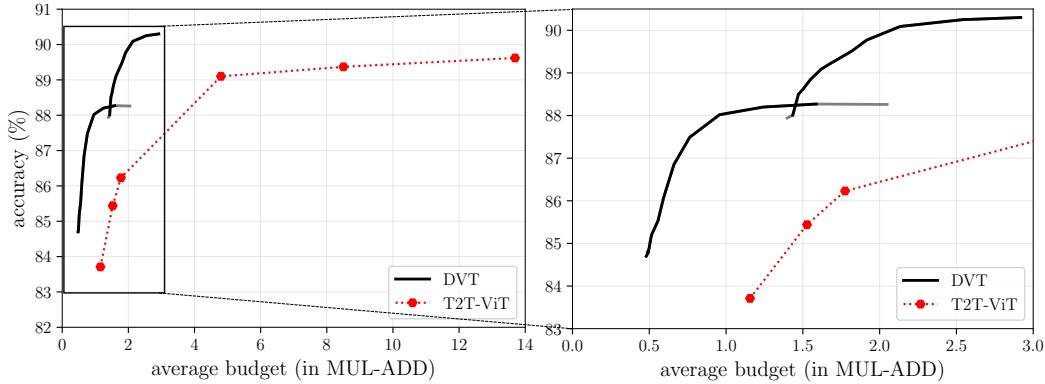

Figure 12: Top-1 accuracy v.s. GFLOPs on CIFAR-100. DVT is implemented on top of T2T-ViT-12/14.

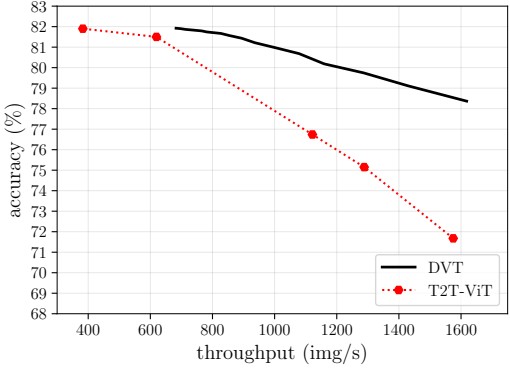

Figure 13: Top-1 accuracy v.s. throughput on ImageNet. The results are obtained on NVIDIA 2080Ti GPU with a batch size of 128.

downstream model. In addition, it is interesting that reusing more relationships only slightly improves the performance. We attribute this to the redundancy within the learned attention logits from different layers of the upstream model.

**Top-1 accuracy v.s. GFLOPs curves on CIFAR-10/100** are presented in Figures 11 and 12, respectively, corresponding to the results reported in Table 3 of the paper.

**Top-1 accuracy v.s. throughput curves on ImageNet** are presented in Figures 13 corresponding to the results reported in Table 2 of the paper.