# OpenReview forum: "Not All Images are Worth 16x16 Words: Dynamic Transformers for Efficient Image Recognition"
_NeurIPS.cc/2021/Conference — NeurIPS 2021 Poster_

### Official Review · Reviewer_kQFQ · 2021-07-06

**Rating:** 7
**Confidence:** 5

**Summary:**

The paper describes how to dynamically apply a cascade of increasingly expensive ViT models in order to save compute resources at inference time. The architecture reuses features and attention logits between tokens of cheaper models as context for the more expensive ones. The resulting models achieve very strong performance on imagenet against competitive baselines when normalized by inference time and compute (flops). Detailed ablations verify individual design choices.

**Limitations And Societal Impact:**

Yes.

**Main Review:**

## Strengths

* strong inference efficiency gains by coarse to fine processing of an image
* clearly written
* detailed ablations
* dynamic model that is able to adapt based on compute budget

## Weaknesses

* No training efficiency gains (yet), actually the proposed model is a bit more expensive because it involves training the entire cascade of models on every image (is this really necessary?). This fact should be made more explicit (!), as training efficiency is quite important, especially because ViTs have shown larger advantages in high-data regimes [1,2]. The paper typically talks about efficiency gains (in general), but should be more precise at times (e.g., table/figure captions, but also occasionally in text).
* No comparison/discussion around other methods that speed up inference (e.g., distillation).
* Only ImageNet.


## Conclusion

Overall I like the idea of coarse to fine processing a lot, and combining it with dynamic processing even more. It goes against the main stream application of pyramids that start processing on the most fine grained level and building up ever more complex features. This is a very wasteful process. I would have liked the paper even more if the authors would have made an effort into translating these ideas also into the training process, which I think would be way more interesting than inference speed-ups, as there are other solutions to this.

I dislike the sole focus on ImageNet (which is unfortunately still common practice) because it is totally unclear to me how these speed ups would translate to other datasets or other tasks. It also makes me question how important some of the design choices are in general as ImageNet1k is fairly small by today's standards. I would urge the authors to at least include results for models pretrained on the freely available ImageNet 21k dataset.

Another important shortcoming of the paper is its lack of comparison to or at least discussion around related work in the space of inference speed-up, e.g. distillation methods.

## Comments & Questions

* capitalize everywhere -> Vision Transformers
* DeiT-small (w/o distillation) --> ViT-small (It is a ViT model with specs introduced in the DEIT paper, but it is still just ViT).
* Why concatenating the low res features before the MLP and why not simply concatenating the low res token reps with the higher res patch embeddings  passing them jointly as input to the transformer? This seems somewhat more elegant to me.

[1] ViT. https://arxiv.org/pdf/2010.11929.pdf
[2] CLIP. https://arxiv.org/pdf/2103.00020.pdf

## UPDATE

I stick with my initial judgement of the paper. I think it deserves acceptance.

**Time Spent Reviewing:**

2

---

> ### Author Response · Authors · 2021-08-10
> **Response to Reviewer kQFQ**
>
>
> Thank you for the valuable suggestions. Our responses to the comments are listed below:
> > **Comment 1:**
> > + No training efficiency gains (yet), actually the proposed model is a bit more expensive because it involves training the entire cascade of models on every image (is this really necessary?). This fact should be made more explicit (!), as training efficiency is quite important, especially because ViTs have shown larger advantages in high-data regimes [1,2]. The paper typically talks about efficiency gains (in general), but should be more precise at times (e.g., table/figure captions, but also occasionally in text).
>
>
> **Reply 1:** \
> Thanks for your valuable suggestion. We agree that improving the training efficiency of ViTs is an important problem. But we would like to argue that the inference efficiency is equally important, if not more. For the latter, there indeed exist some other solutions (e.g., distillation and pruning). However, we believe that dynamic inference is still a valuable research direction. As we have empirically shown below (see: **Reply 2**), DVT achieves competitive performance compared with some of the representative baselines. Moreover, techniques like distillation and pruning are complementary to our method as they can be implemented on top of DVT to further improve efficiency.
>
>
> Concerning the training efficiency, it is possible to perform early-exiting on DVT *during training*. In fact, we have preliminarily tested this idea, but so far we haven't obtained quite positive results that worth being reported. We are happy to further explore this idea, and will include discussions on this in our revision.
>
>
> > **Comment 2:**
> > + No comparison/discussion around other methods that speed up inference (e.g., distillation).
>
>
> **Reply 2:** \
> Thanks for the suggestion. We compare our proposed DVT with both knowledge distillation (KD) [*1] and patch pruning based methods [*2, *3] in the following three tables. One can observe that DVT outperforms all the baselines with similar FLOPs, or has significantly lower computational cost with the same accuracy. Besides, one can expect that DVT is compatible with these techniques, i.e., DVT can also be trained with KD to further boost the accuracy or leverage patch pruning to reduce the FLOPs. Both the results and discussions will be included in our revision.
>
> ||GFLOPs/image|ImageNet Top-1 acc.|
> |-----|-----|-----|
> |DeiT-small|4.61|79.80%|
> |DeiT-small + distillation [*1]|4.63|81.20%|
> |DVT (DeiT-small)|4.63|**81.67%**|
> |DVT (DeiT-small)|**3.61**|81.20%|
>
> ||GFLOPs/image|ImageNet Top-1 acc.|
> |-----|-----|-----|
> |DeiT-small|4.61|79.80%|
> |DeiT-small + DynamicViT [*2]|2.90|79.30%|
> |DVT (DeiT-small)|2.90|**80.40%**|
> |DVT (DeiT-small)|**2.37**|79.30%|
>
> ||GFLOPs/image|ImageNet Top-1 acc.|
> |-----|-----|-----|
> |DeiT-small|4.61|79.80%|
> |DeiT-small + Patch Slimming [*3]|2.60|79.40%|
> |DVT (DeiT-small)|2.60|**79.81%**|
> |DVT (DeiT-small)|**2.41**|79.40%|
>
>
> [*1] Touvron, Hugo, et al. "Training data-efficient image transformers & distillation through attention." International Conference on Machine Learning. PMLR, 2021. \
> [*2] Rao, Yongming, et al. "DynamicViT: Efficient Vision Transformers with Dynamic Token Sparsification." arXiv preprint arXiv:2106.02034 (2021). \
> [*3] Tang, Yehui, et al. "Patch Slimming for Efficient Vision Transformers." arXiv preprint arXiv:2106.02852 (2021).
>
>
> > **Comment 3:**
> > + Only ImageNet.
> >
>
> **Reply 3:** \
> We agree that reporting the results on ImageNet-21k will make our experiments stronger, and we are happy to follow your advice. However, due to limited time and computational resources, the ImageNet-21k results may be added in our revision after rebuttal. We plan to pre-train DVT (T2T-ViT-14) and DVT (T2T-ViT-19) following the hyper-parameters of Swin Transformer [*5].
>
> Still, we believe that our current results on ImageNet-1k are representative to demonstrate the effectiveness of DVT. Previous observations [*4, *5, *6] have shown that higher ImageNet-1k accuracy usually translates into better performance on downstream tasks (e.g, on Oxford Flowers, CIFAR-10/100, ADE20K and MS COCO).
>
> [*4] Dosovitskiy, Alexey, et al. "An Image is Worth 16x16 Words: Transformers for Image Recognition at Scale." International Conference on Learning Representations. 2020. \
> [*5] Liu, Ze, et al. "Swin transformer: Hierarchical vision transformer using shifted windows." arXiv preprint arXiv:2103.14030 (2021). \
> [*6] Wang, Wenhai, et al. "Pyramid vision transformer: A versatile backbone for dense prediction without convolutions." arXiv preprint arXiv:2102.12122 (2021).
>
>
>
> > **Comment 4:**
> > + capitalize everywhere -> Vision Transformers
>
> **Reply 4:** \
> Thanks, we have carefully fixed these issues.
>
>
>
> > **Comment 5:**
> > + DeiT-small (w/o distillation) --> ViT-small (It is a ViT model with specs introduced in the DEIT paper, but it is still just ViT).
>
> **Reply 5:** \
> Indeed, the network architecture of DeiT-small (w/o distillation) is the same as ViT-small. However, since DeiT introduces a more advanced training strategy (see: Table 9 in their paper), they are not totally identical. We will make these clear in revision.
>
> > **Comment 6:**
> > + Why concatenating the low res features before the MLP and why not simply concatenating the low res token reps with the higher res patch embeddings passing them jointly as input to the transformer? This seems somewhat more elegant to me.
>
> **Reply 6:** \
> This is a clean idea. However, a potential issue is that it may result in significantly increased computational cost in downstream Transformers due to the increased token number. Take DVT (T2T-ViT-12) for example (see: Table 4 of the paper). With this idea, the numbers of representative tokens in the $2^{\text{nd}}$ and $3^{\text{rd}}$ Transformers will be increased by $100 \to 100+49$ and $196 \to 196+100+49$, respectively. Correspondingly, the FLOPs of the $3^{\text{rd}}$ exit will be increased by 57.5%. In comparison, our reuse mechanisms only lead to 11.1% more FLOPs. Some improvements may be needed to alleviate this problem.

---

> > ### Comment · Reviewer_kQFQ · 2021-08-16
> > **RE**
> >
> > Thanks for the responses.
> >
> > Re Reply 6:
> >
> > Compute will be more expensive (though less than 2x), yes, but the question is how much do you get out of the additional compute. It could be that it is well worth it when plotting on performance/compute.

---

> > > ### Author Response · Authors · 2021-08-22
> > > **Re: Re: Reply 6**
> > >
> > > Thank you for the response. We agree that such a clean idea is definitely worth exploring. In fact, we are now trying to implement it on the basis of the three-exit DVT (T2T-ViT-12) to see whether it will lead to better computational efficiency. Once positive results are obtained, we are happy to include them in appendix and further explore this idea in the future.

---

### Official Review · Reviewer_TPUT · 2021-07-17

**Rating:** 6
**Confidence:** 4

**Summary:**

This paper introduces an efficient inference strategy for vision transformers (ViTs). The motivation is that there are some “easy” images which can be classified using a few tokens. Based on the motivation, the paper proposes a cascading inference framework which do not proceed to next stage if the prediction confidence is enough high (i.e., early exit  strategy). For further efficiency of the cascading scheme, the previous stage’s features are fused with the next stage features. Experimental results show that the proposed method outperforms the baseline methods both in inference speed and accuracy.


**Limitations And Societal Impact:**

Yes.

**Main Review:**

### Strengths
- This paper is well-written and the motivation is clear.
- Experiments are extensive with various datasets and plenty of ablation studies.


### Concerns and questions
- The proposed method provides technical contributions to accelerate vision transformers with an interesting motivation. However, the early-exiting is a  well-known strategy in CNN fields as the authors mentioned in the related work section ([21, 46, 24]).
  - It would be nice to see in-detail comparisons against the existing multi-stage early-exiting strategies [21, 46, 24] in both empirical and conceptual ways.
  - Could the previous methods [21, 46, 24] be naively adopted to vision transformer architectures? If not, discussing the reasons would be very interesting.
- The proposed framework reuses the previous stage’s features to enhance the accuracy. The “reuse” mechanism is quite interesting, so it would be great if the paper provides a discussion about “reusing” the previous features to improve the inference speed of the next stages.

### Post-rebuttal comments
- The author's response (reply) has resolved most concerns I raised so that I will keep my initial rating (weak acceptance).



**Time Spent Reviewing:**

6

---

> ### Author Response · Authors · 2021-08-10
> **Response to Reviewer TPUT**
>
>
>
> Thank you for the valuable suggestions. Our responses to the comments are listed below:
> > **Comment 1:**
> > + It would be nice to see in-detail comparisons against the existing multi-stage early-exiting strategies [21, 46, 24] in both empirical and conceptual ways.
>
> **Reply 1:** \
> Thank you for your comment. The empirical and conceptual comparisons are presented below. Both the new results and discussions will be included in our revision.\
> (1) *Empirical comparisons.* We empirically compare the proposed DVT and existing CNN-based early-exiting networks [21, 24, 46] in the following table. The Top-1 accuracy on ImageNet under different computational budgets is reported. One can observe that DVT significantly outperforms these baselines.
>
> |GFLOPs/image|0.75|1.0|1.25|1.5|1.75|2.0|
> |-----|-----|-----|-----|-----|-----|-----|
> |MSDNet [21]|69.82%|71.24%|72.73%|73.66%|73.99%|74.20%|
> |IMTA-MSDNet [24]|70.84%|71.92%|73.41%|74.31%|74.64%|74.94%|
> |RANet [46]|70.48%|72.24%|73.57%|74.56%|75.02%|75.10%|
> |DVT (T2T-ViT-12)|**73.68%**|**76.19%**|**77.84%**|**78.87%**|**79.47%**|**79.75%**|
>
> (2) *Conceptual comparisons.* A major efficiency gain in both MSDNet [21] and RANet [46] comes from the adaptive depth, namely processing 'easier' samples using fewer layers within a *single* network. In comparison, DVT cascades multiple Transformers to infer several *entire* networks at test time (comparisons of the two approaches can be found in **Reply 2**). Besides, DVT does not leverage the *multi-scale architecture* or *dense connection* in MSDNet [21]. Compared with RANet [46], DVT introduces the *adaptive token number* and the *relationship reuse mechanism* tailored for ViT. In addition, IMTA [24] studies the training techniques of multi-exit models, which are actually complementary to DVT. Another difference is that all the three works [21, 24, 46] adopt *pure convolutional models*.
>
>
> > **Comment 2:**
> > + Could the previous methods [21, 46, 24] be naively adopted to vision transformer architectures? If not, discussing the reasons would be very interesting.
>
> **Reply 2:** \
> We test applying the adaptive depth paradigm [21, 46] to ViT in the following table. Two classifiers are uniformly added to the intermediate layers of T2T-ViT-14, and the early-exiting is performed with the three exits in the same way as DVT. One can observe that the layer-adaptive T2T-ViT achieves significantly lower accuracy than DVT using identical computational budgets. This inferior performance may be alleviated by improving the network architecture, as shown in [21]. We are happy to explore this in the future.
>
> |GFLOPs/image|2.0|2.5|3.5|4.5|4.0|
> |-----|-----|-----|-----|-----|-----|
> |T2T-ViT-14 with adaptive depth|76.09%|78.88%|79.46%|79.54%|79.55%|
> |DVT (T2T-ViT-14)|**79.24%**|**80.61%**|**81.47%**|**82.10%**|**82.42%**|
>
> In addition, it is worth noting that both the aforementioned adaptive depth mechanism [21, 46] and the training techniques proposed in [24] are actually complementary to DVT. It is possible to add intermediate classifiers to any one of the cascaded Transformers in DVT, and train all the exits of DVT following [24]. The idea of adaptive resolution [46] is tailored for convolutional models, where the computational cost grows with respect to the image height/width. In contrast, the FLOPs of Transformers are directly related to the token number rather than the resolution, and hence it is more reasonable to consider dynamic tokens, as we do in DVT.
>
>
>
>
> > **Comment 3:**
> > + The proposed framework reuses the previous stage’s features to enhance the accuracy. The “reuse” mechanism is quite interesting, so it would be great if the paper provides a discussion about “reusing” the previous features to improve the inference speed of the next stages.
>
> **Reply 3:** \
> Since there is a trade-off between accuracy and speed, it is totally possible to reduce the computational cost of downstream Transforms with the reuse mechanisms. To see this, we test reducing the size of the $2^{\text{nd}}$ Transformer on the basis of the two-exit DVT we use in the ablation study of feature reuse (see: Table 5 of the paper). Two variants are considered, i.e., reducing the dimension of tokens and reducing the number of layers. The computational cost (includes reuse) and accuracy of the $2^{\text{nd}}$ Transformer are shown in the following table. It can be observed that the feature reuse mechanism can reduce the GFLOPs by up to $\sim$30% with even higher accuracy than the baseline. More results will be added in our revision.
>
> ||GFLOPs of the $2^{\text{nd}}$ Transformer|Top-1 acc. of the $2^{\text{nd}}$ Transformer|
> |-----|-----|-----|
> |Baseline (w/o reuse, token dim=256, layer num=12)|0.90|73.61%|
> |w/ reuse, layer number: 12 $\to$ 10|0.83 ($\downarrow$ 7.8%)|74.68% ($\uparrow$ 1.07%)|
> |w/ reuse, layer number: 12 $\to$ 8|0.70 ($\downarrow$ 22.2%)|73.80% ($\uparrow$ 0.19%)|
> |w/ reuse, token dim: 256 $\to$ 224|0.79 ($\downarrow$ 12.2%)|74.58% ($\uparrow$ 0.97%)|
> |w/ reuse, token dim: 256 $\to$ 192|0.65 ($\downarrow$ 27.8%)|74.11% ($\uparrow$ 0.50%)|

---

### Official Review · Reviewer_SzLL · 2021-07-19

**Rating:** 6
**Confidence:** 4

**Summary:**

This paper proposes a dynamic transformer network that allocate different computational cost to process different images with different complexity. The key idea is to change the number of tokens to represent an images, and based on the output confidence, the model decides if it goes to the next stage of computation to get more confident results. This paper further proposes two "reuse" strategies to reuse the results from the previous stage. Experiments show this improves the performance.

**Limitations And Societal Impact:**

No societal impact discussion needed, in my opinion.

**Main Review:**

Strength:
+ This paper successfully applied the mutl-stage (also called early exit, dynamic models, etc.) strategies to improve the efficiency of Vision Transformers.
+ One unique contribution of this paper is that it tries to reuse computation from previous stages and use them to improve the performance. Unique to ViT, this paper proposes "feature reuse" and "relationship reuse".

Weakness:
- The early-exit idea has been widely adopted in CNN models. The high-level idea of this paper is not that different from early-exit, though it is now applied on transformers. The novelty of this work is therefore limited.

- Since the computation of such multi-stage network is inherently serial, the latency of such networks may be slow even though the theoretical FLOPs are low.

- Different samples can have different latencies, this can be problematic for applications with hard latency constraint. Also, the batch-processing of such method can be much more complicated.

Question:
How is the actual throughput/latency measured?  I assume in this paper, it reported the "average" latency/throughput. Since the speed is measured on GPUs with batch processing, if one sample is more "difficult" and requires multiple stages of processing, how is the latency of this batch calculated? Assuming that a sample's latency is t_{b,i}, where b is the batch index and "I" is the index within the batch. The time it takes to finish inference of this batch should be max_i t_{b,i}, which means that even there are many simple (fast) samples, the inference time for this batch is determined by the slowest sample's processing time. I hope the author can clarify how their speed is measured, and how to avoid the problem I mentioned above.

**Time Spent Reviewing:**

1

---

> ### Author Response · Authors · 2021-08-10
> **Response to Reviewer SzLL**
>
>
> Thank you for the valuable suggestions. Our responses to the comments are listed below:
>
> > **Comment 1:**
> > How is the actual throughput/latency measured?
>
> **Reply 1:** \
> We consider the scenario where a large batch of test samples need to be processed. In specific, the 50,000 images in the validation set of ImageNet are fed into the proposed DVT with a batch size of 128 on a single NVIDIA 2080ti GPU. At each mini-batch, the samples that meet the early-termination criterion of a certain exit will be output, with the remaining images continued to be processed by the downstream Transformer. The throughput is computed by 50,000/$T$, where $T$ is the total wall-clock time of processing all the mini-batches.
>
> We will make this procedure more clear in our revision.
>
>
> > **Comment 2:**
> > Assuming that a sample's latency is t_{b,i}, where b is the batch index and "I" is the index within the batch. The time it takes to finish inference of this batch should be max_i t_{b,i}, which means that even there are many simple (fast) samples, the inference time for this batch is determined by the slowest sample's processing time.
>
> **Reply 2:** \
> Thanks for the comment. In fact, what you mentioned holds true *only if* the batch is very small or the computational resource is sufficiently abundant. While in a more practical setting, the case is different.
>
> To see this, we first define a "computation unit  (CU)" as the minimum hardware resource (e.g., a set of GPU cores) that need to process a single image in parallel. With this definition, the time a CU needs to process $k$ samples is $k$ times the latency of processing a single sample.
>
> Suppose that we have a batch of $N$ samples and $M$ CUs. If $N \leq M$, all samples can be processed in parallel (each sample is processed with one CU), then the latency is determined by the slowest sample's inference time, which is the case you pointed out.
>
> However, in a more practical setting where we have a very large batch of samples that need to be inferred, we can always ensure that $N \gg M$ (we would put a sufficiently large batch of samples on each GPU as a standard of practice). Consequently, each CU needs to process $N/M$ samples. It is easy to ensure that samples of different latency are uniformly distributed on different CUs, thus all the CUs are not idle before the batch has been processed. Therefore, the latency of the batch is $(\sum_{i=1}^N t_i)/M$, instead of $\max_{i=1,\ldots, N} \{t_i\}$, where $t_i$ is the latency for the $i$ th sample.
>
>
> > **Comment 3:**
> > The early-exit idea has been widely adopted in CNN models. The high-level idea of this paper is not that different from early-exit, though it is now applied on transformers. The novelty of this work is therefore limited.
>
> **Reply 3:** \
> We would like to argue that our novelty mainly lies in that we for the first time proposed the idea of *automatically configuring the token numbers* in Vision Transformers (ViT). Although we adopted the widely used early-exit idea, our work is not a trivial extension of existing approaches. To optimize the efficiency of the model, we have designed *novel feature/relationship reuse* modules tailored for ViTs to reduce redundant computations across different exits. We have empirically shown that the two modules are critical for our DVT to achieve competitive performance.
>
>
>
>
> > **Comment 4:**
> > Since the computation of such multi-stage network is inherently serial, the latency of such networks may be slow even though the theoretical FLOPs are low.
>
> **Reply 4:** \
> As we discuss in **Reply 2**, the latency of DVT *can* match its theoretical FLOPs under a practical batch inference setting. This has been empirically validated by our results on actual inference speed (i.e., Table 2 in the paper). DVT significantly improves the efficiency in practice even though a multi-stage architecture is adopted.
>
> Moreover, on mobile computing platforms where the task is usually inherently serial, it is more straightforward for DVT to realize its theoretical efficiency.
>
>
> > **Comment 5:**
> > Different samples can have different latencies, this can be problematic for applications with hard latency constraint.
>
> **Reply 5:** \
> In fact, we can ensure that DVT meets a given hard latency constraint by design, e.g., simply restricting the FLOPs/latency of its last exit  (i.e., the maximum possible FLOPs/latency) under that constraint.
> Note that a *static* evaluation of DVT (i.e., w/o early-exiting samples) yields comparable performance as ViT of similar size, as shown in the following two tables. Based on this fact, when we allow a dynamic evaluation of DVT, redundant computation can be saved for many 'easy' images without sacrificing accuracy.
>
> Although the latency can not be reduced for *all* the samples, DVT is valuable as it accelerates the inference most of the time, and reduces the overall power consumption.
>
> ||GFLOPs/image|ImageNet Top-1 acc.|
> |-----|-----|-----|
> |DeiT-base|17.58|81.80%|
> |The $3^{\text{rd}}$ exit of DVT (DeiT-small)|**8.84**|**81.81%**|
>
> ||GFLOPs/image|ImageNet Top-1 acc.|
> |-----|-----|-----|
> |T2T-ViT-24|13.69|82.30%|
> |The $3^{\text{rd}}$ exit of DVT (T2T-ViT-14)|**9.33**|**82.81%**|
>
>
>
> > **Comment 6:**
> > Also, the batch-processing of such method can be much more complicated.
>
> **Reply 6:** \
> In fact, the batch-processing of DVT is *not complicated*. As discussed in **Reply 1**, one can simply output the samples that meet the early-termination criterion at each exit, and feed the remaining images in the batch into the downstream Transformer. The former can be achieved with a comparison operation (prediction confidence v.s. threshold), while the latter only needs to index the remaining samples. Our code and models will be released.

---

> > ### Comment · Reviewer_SzLL · 2021-08-31
> > **RE**
> >
> > Thanks for the clarification. I think the paper would be stronger if you would put the latency discussion (Reply 2) in the paper, so the community can have a better understanding of when to use this method.
> >
> > > It is easy to ensure that samples of different latency are uniformly distributed on different CUs
> >
> > This may not be always true, depending on the input data's distribution. Assuming the input data are i.i.d., then I assume a random assignment could satisfy. But in certain scenarios, say, when the input are temporally dependent and correlated, it is possible that images assigned to certain CUs are all hard or all easy. It might be useful to come up with more advanced solutions to ensure the uniform distribution of difficulties.
> >
> > Overall, I think the author addressed my questions and I will hold my initial rating to weak accept this paper.

---

### Decision · Program_Chairs · 2021-09-27

**Decision:**

Accept (Poster)

**Comment:**

All three reviewers are positive about the paper (also after considering the authors' response and discussion). Here are the main points:

Pros:
1) Well motivated and well presented idea
2) Good inference speed improvements on ImageNet
3) Good ablation studies

Cons:
1) Somewhat related to existing early-exiting strategies
2) Doubts about practical gains in inference/training speed

The authors were successful at addressing the reviewers' concerns. Overall, the idea is clean, is presented well, and works well. Thus I recommend the acceptance of the paper.